# Oxidative Stress by H_2_O_2_ as a Potential Inductor in the Switch from Commensal to Pathogen in Oncogenic Bacterium *Fusobacterium nucleatum*

**DOI:** 10.3390/antiox14030323

**Published:** 2025-03-07

**Authors:** Alessandra Scano, Sara Fais, Giuliana Ciappina, Martina Genovese, Barbara Granata, Monica Montopoli, Pierluigi Consolo, Patrizia Carroccio, Paola Muscolino, Alessandro Ottaiano, Alessia Bignucolo, Antonio Picone, Enrica Toscano, Germano Orrù, Massimiliano Berretta

**Affiliations:** 1Oral Biotechnology Laboratory (OBL), Department of Surgical Science, University of Cagliari, 09124 Cagliari, Italy; alessandra.scano77@unica.it (A.S.); sarafais79@gmail.com (S.F.); 2Molecular Biology Laboratory, Azienda Ospedaliera Universitaria di Cagliari (AOU), 09124 Cagliari, Italy; 3Division of Medical Oncology, AOU “G. Martino” Hospital, University of Messina, 98124 Messina, Italy; giulianaciappina@gmail.com (G.C.); antonio.picone@polime.it (A.P.); 4Department of Clinical and Experimental Medicine, University of Messina, 98122 Messina, Italy; martinagenovese1999@gmail.com (M.G.); pierluigi.consolo@unime.it (P.C.); abignucolo@unime.it (A.B.); 5School of Specialization in Medical Oncology, Department of Human Pathology “G. Barresi”, University of Messina, 98125 Messina, Italy; grnbbr97c64c351c@studenti.unime.it (B.G.); patrizia.carroccio@studenti.unime.it (P.C.); paola.muscolino@studenti.unime.it (P.M.); tscnrc94t62f158j@studenti.unime.it (E.T.); 6Department of Pharmaceutical and Pharmacological Sciences, University of Padova, 35122 Padua, Italy; monica.montopoli@unipd.it; 7Division of Innovative Therapies for Abdominal Metastases, Istituto Nazionale Tumori IRCCS Fondazione G. Pascale, 80131 Naples, Italy; a.ottaiano@istitutotumori.na.it

**Keywords:** *Fusobacterium nucleatum*, colon-rectal cancer, oxidative stress, pri-miRNA, gene expression

## Abstract

Background: *Fusobacterium nucleatum* is a pathobiont that plays a dual role as both a commensal and a pathogen. The oral cavity typically harbors this anaerobic, Gram-negative bacterium. At the same time, it is closely linked to colorectal cancer due to its potential involvement in tumor progression and resistance to chemotherapy. The mechanism by which it transforms from a commensal to a pathogen remains unknown. For this reason, we investigated the role of oxidative status as an initiatory factor in changing the bacterium’s pathogenicity profile. Methods: A clinical strain of *F. nucleatum* subsp. *animalis* biofilm was exposed to different oxidative stress levels through varying subinhibitory amounts of H_2_O_2_. Subsequently, we investigated the bacterium’s behavior in vitro by infecting the HT-29 cell line. We evaluated bacterial colonization, volatile sulfur compounds production, and the infected cell’s oxidative status by analyzing *HMOX1*, *pri-miRNA 155*, and *146a* gene expression. Results: The bacterial colonization rate, dimethyl sulfide production, and *pri-miRNA 155* levels all increased when stressed bacteria were used, suggesting a predominant pathogenic function of these strains. Conclusions: The response of *F. nucleatum* to different oxidative conditions could potentially explain the increase in its pathogenic traits and the existence of environmental factors that may trigger the bacterium’s pathogenicity and virulence.

## 1. Introduction

The oral Gram-negative bacterium *Fusobacterium nucleatum* is one of the most studied and cited microbial species in the field of oral medicine, as well as in various systemic conditions, with particular attention given to its role as an onco-bacterium [1]. Traditionally, it was primarily associated with periodontal diseases and described by Sokransky et al. in the orange microbial complex [2]. For several decades, this microorganism’s role as an oral periodontal bacterium was emphasized, even though the related periodontal diseases were clinically associated with other systemic illnesses such as cardiovascular diseases, preterm birth, and rheumatoid arthritis [3,4,5]. Several key articles published in the early 2000s viewed this bacterial species as an opportunistic pathogen. Jern Aas’s group used next-generation sequencing (NGS) to analyze the *16S rRNA* gene profile in nine oral sites from five healthy individuals [6]. The study revealed the presence of 141 predominant species, more than 60% of which were non-cultivable bacteria. Interestingly, samples from the main areas of the mouth—such as the front part of the upper jaw, the back and side surfaces of the tongue, the hard palate, and the surfaces of the teeth—tested positive for *Fusobacterium* spp., with *F. nucleatum* subsp. *polymorphum*, and *F. nucleatum* subsp. *animalis* being the most representative subspecies. Additionally, Starkenmann et al. (2008) contributed to uncovering a physiological role of *F. nucleatum* in the olfactory perception of different edible vegetables [7,8]. The natural presence of this bacterium in healthy individuals, along with its proven physiological function, challenges the traditional paradigm that equates pathogens with disease. Subsequently, various researchers have highlighted the dual role of *F. nucleatum* as both a commensal and a pathogen [1,8]. Recent studies suggest that a significant shift in the virulence of *F. nucleatum* likely arises from alterations in the oral microflora. This shift manifests as a change in the relative abundance of various microbial groups, which also leads to a significant alteration in the pathogenicity profile of some of these organisms. Clinical reports often associate this shift with oral dysbiosis, frequently manifesting as symptoms such as halitosis [9]. Thus, an interesting question arises: what host condition or intrinsic biological factor triggers *F. nucleatum* to transform into a pathogen? According to the existing literature, one common biological process involved is the oxidative status of host tissues, a condition in the oral cavity closely related to host habits and other factors influencing oral pathophysiology [10,11,12]. Some studies have investigated how the bacterium’s proteome changes during an oxidative burst and found that three major protein systems become more active in the *F. nucleatum* cytoplasm. The decrease in ATP levels and the increase in intracellular concentrations of chaperone-type proteins drive this increase in stress-related proteins, a process closely linked to oxygen availability. Furthermore, the partial tolerance of *F. nucleatum* to aerobic conditions may enhance its virulence, an aspect observed using techniques such as ESI-Q/TOF-MS. Research using tumor cell lines has also shown that infection under hypoxic conditions may amplify malignant transformation [13,14,15,16,17,18]. Moreover, several studies have demonstrated how *F. nucleatum* contributes to adenoma/carcinoma conversion by promoting the nuclear translocation of β-catenin and the overexpression of the regenerative gene (REG) *Iα*, which codes for proteins that have proliferative and anti-apoptotic effects on inflammatory or neoplastic lesions in the gastrointestinal tract [19,20,21,22]. Additionally, a higher amount of *F. nucleatum* in colorectal cancer (CRC) tissue has been correlated with shorter survival rates [23] and higher recurrence post-chemotherapy [24]. FadA, Fap2, and endotoxin LPS are the three main virulence factors found in *F. nucleatum*. Furthermore, many articles often include outer membrane proteins RadD, CmpA, FomA, serine proteases, and butyric acid. These proteins work together to make an environment conducive to cancer growth, and they affect both the immune system and the oxidative stress pattern in cells in a big way.

However, it remains unclear how *F. nucleatum* contributes to the development of its pathogenic features. In this study, we hypothesize that a set of “preparatory stressing conditions” already present in the oral cavity may alter the host–microbe interaction profile of *F. nucleatum*. This activated bacterium could then have the potential to invade colorectal cells via the oral–gut axis with enhanced capacity [14]. To investigate this, we present an in vitro model to measure cell colonization rates and stress signals in HT-29 cell lines infected with stress-activated strains of *F. nucleatum.*

## 2. Materials and Methods

### 2.1. Strain Used in This Work

For this study, we used the clinical isolate of *F. nucleatum* subsp. *animalis* CA-100. The bacterium was cultured on Columbia agar base (DIFCO^TM^, Becton, Dickinson, Sparks, MD, USA) and maintained under strict anaerobic conditions in a jar for 4 days using the AnaeroJar/AnaeroGen system (Thermo Scientific™ Oxoid™, Waltham, MA, USA). Strain identification was confirmed by Sanger DNA sequencing of the 16S rRNA gene region. Using the BLAST program (https://blast.ncbi.nlm.nih.gov/Blast.cgi, version 09/24, accessed on 28 February 2025), the obtained nucleotide sequence matched perfectly with GenBank accession number CP077184.1 (strain SB030). The representative *F. nucleatum* colonies were transferred to brain–heart infusion (BHI) liquid medium (Microbiol-Uta Cagliari, Italy) supplemented with 2.5 g of yeast extract, 0.3 g of L-cysteine, 5 µg/mL of hemin, and 1 µg/mL of menadione (Sigma-Aldrich, Burlington, MA, USA) [15], which had been deoxygenated. The liquid culture was placed in an anaerobic atmosphere (bags from Sigma Aldrich, Burlington, MA, USA) and maintained until it reached the mid-exponential growth phase, which was evaluated at 600 nm with an absorbance of 0.55 after approximately 17 h, using a spectrophotometer (Sequoia-Turner 690, CA, USA, optical path length = 10 mm). For the *F. nucleatum* stress experiment, we used this medium, which included BHI added with 20% human saliva from a healthy subject, previously sterilized by filtration 0.45 μm, A-Medium (Figure 1). A strain of *Escherichia coli* ATCC 7075 (American Type Culture Collection) was used as the control strain in the biofilm experiment. This strain was cultured in BHI broth at 37 °C and stored at −80 °C in BHI with 20% glycerol (Merck KGaA, Darmstadt, Germany) until use.

### 2.2. Antibacterial Activity of H_2_O_2_

An antimicrobial susceptibility test was used to find the range of concentrations of H_2_O_2_ that would not inhibit the growth of the *F. nucleatum* strain that was used in this study. The procedure was based on a modified method that Orrù et al. have already described for other bacterial strains [16]. 1 × 10^6^ CFU/mL cells were put into a Nunc 96 Well (Thermo Fisher Scientific, Waltham, MA, USA) microplate that had 200 μL of medium A and varying concentrations of hydrogen peroxide in each well, ranging from 1 × 10^5^ to 125 µM. The absorbance was measured at 550 nm after 48 h of incubation in anaerobic conditions at 37 °C [16]. After 48 h of incubation at 37 °C in an anaerobic atmosphere (bag system, Merck KGaA, Darmstadt, Germany), the absorbance was measured at 550 nm by the Multiskan™ FC Microplate Photometer (Fisher Scientific, Milan, Italy). Non-active concentrations of H_2_O_2_ were those that showed the absorbance equal to the positive control. In this context, we evaluated the minimum inhibitory concentration (MIC), as the lowest H_2_O_2_ concentration that prevented the growth of *F. nucleatum*.

### 2.3. Stressing-Chamber and VSC Measurement

In accordance with the concept that a chemical stress can trigger a distinct virulence pattern in *F. nucleatum*, we have subjected a culture of this strain to stress using a specific stressing chamber prior to the cellular infection. Additionally, measuring volatile sulfur compounds (mainly H_2_S and CH_3_SH, (CH_3_)_2_S) in the air inside the chamber has a direct connection to the activity of bacterial proteases and then, with its pathogenicity. As illustrated in Figure 1, the stressing chamber operated under anaerobic conditions, with an atmosphere of 5% H_2_, 5% CO_2_, and 90% N_2_. Specifically, *F. nucleatum* was structured as a biofilm within a scaffold composed of natural cotton [17] inside a tube as shown in the Figure 1. After 72 h of incubation (ABS 0.130 at λ = 550 nm), the medium (Medium-A) was dismissed and the scaffold containing biofilm bacteria was submerged in human sterile saliva. The stress was applied by adding different concentrations of hydrogen peroxide to the biofilm surfaces, from 2000 µM to 10 µM. After 24 h of incubation, the VSC production was measured in the air of each tube. This was performed to test the bacterium protease activity by VSC analysis using gas chromatography with the Oral Chroma™ apparatus (Abilit, Tokyo, Japan) [18]. The *F. nucleatum* isolate shown to have the highest VSC activity was used as test in the 7H9 cell infection experiment.

### 2.4. Cells Used in This Study

We used HT-29 human colon adenocarcinoma cells ATCC (American Type Culture Collection). This cell line serves as a suitable model for infection, finding applications in both cancer and toxicology research [19,20]. The cells were cultured on RPMI 1640 medium (Gibco^TM^, Thermo Fisher Scientific, Waltham, MA, USA) supplemented with 2% L-glutamine, 10% fetal bovine serum, and 1% penicillin/streptomycin (Sigma-Aldrich, Burlington, MA, USA). The cells were then placed in an incubator in a humidified atmosphere with 5% CO_2_ at a temperature of 37 °C.

### 2.5. F. nucleatum Infection Process

We carried out the infection of HT-29 cells with *F. nucleatum* using already described modified protocol [15]. The infection procedure was performed in HT-29 cell culture grown in 6-well plates to 80% confluency, approximately 1 × 10^6^ cells. A stressed strain of *F. nucleatum* was added at different multiplicities of infection (MOI) ratios (from 1/10 to 1/1000). Following the results of Silva et al., *F. nucleatum* remained viable for up to 60 h of molecular oxygen exposure; the infected culture was maintained in an 5% CO_2_ atmosphere inside this range [10]. For each MOI condition, at least three independent infection experiments were performed.

### 2.6. Enumeration of F. nucleatum Cells by Real-Time qPCR

Real-time PCR was performed to quantify the total bacterial genomes in infected HT-29 cells previously detached with 0.05% trypsin-EDTA solution. For DNA extraction, we used the GeneProof DNA Isolation Kit (Gene Proof, Doln Herpice CZ-619 00, Brno, Czech Republic) following the manufacturer’s instructions. The RT-qPCR reaction mix was prepared by using a SYBR Premix Ex Taq Kit (TaKaRa-Clontech^®^, Kusatsu, Japan). The PCR primers used in this work were OG41: (3′-GGCCACAAGGGGACTGAGACA-5′) and OG 42: (3′-TTTAGCCGTCACTTCTTCTGTTGG-5′) [18]. The 0.02 mL final volume contained 1XPremix Ex Taq (2X), 1X SYBR Green (10,000X), 0.22 μM of each primer, and 1 to 10 ng of DNA extract. The qPCR thermal profiles consisted of a denaturation step at 95 °C for 30 s, followed by 40 cycles of 5 s at 95 °C, 30 s at 60 °C, and 20 s at 80 °C. Fluorescence was detected at the end of the 80 °C segment in the PCR step. The bacterium concentration was obtained by interpolating the sample threshold cycle value with the standard curve, which we obtained using a series of *F. nucleatum* cell dilutions, ranging from 5 × 10^2^ to 1 × 10^8^ bacterial genomes/µL.

### 2.7. Oxidative Pattern Evaluation in HT-29 Infected Cells

A set of different molecular markers was used to evaluate the oxidative stress pattern during HT-29 cell *F. nucleatum* infection. Table 1 shows the target genes and respective primers used for real-time RT-qPCR, with 10 μM in PCR mix.

After the infection process, we froze the HT-29 cells in liquid nitrogen and then added 1 mL of TRIzol™ Reagent (Thermo Fisher Scientific Inc.). After gently scraping the plate bottom, we used the obtained suspension for RNA extraction, following the instructions provided in the manufacturer’s instructions. Prior to the RT-PCR reaction, the RNA quality and amount were assessed using a NanoDrop Microvolume Spectrophotometers (Thermo Fisher Scientific Inc.). Gene expression was performed with a RT-qPCR procedure using the relative quantitation approach by 2^−∆∆Ct^ method [21]. The reaction was carried out with SuperScript IV UniPrime One-Step RT-PCR System (Thermo Fisher Scientific Inc.), following the manufacture instruction. Briefly, 20 µL of RT-PCR mixture contained 10 µL of UniPrime™ RT-PCR Master Mix; 2 µL of primer mix, 10 µM each; 0.8 µL SuperScript™ IV RT Mix; 5 µL RNA extract; 2 µL of SYBR Green I (Sigma-Aldrich, Burlington, MA, USA); and 1.2 µL of DNase/RNase free distilled water. RT-PCR was performed by using MX 96 instrument (Bio-Rad Laboratories, Inc., Hercules, CA, USA). The PCR cycles were composed of (i) 10 min at 50 °C, 2 min 98 °C for cDNA synthesis and (ii) 40 cycles of 10 s at 98 °C, 10 s at 60 °C, and 30 s at 72 °C. Fluorescence was measured at the end of the 60 °C step.

### 2.8. F. nucleatum Biofilm Measurement

The ability of *F. nucleatum* to form biofilm in the presence of different concentrations of hydrogen peroxide was evaluated using the crystal violet staining procedure, described by Aragoni et al. (2023) [22]. The experiment was conducted in a 96-well microplate, containing 200 µL per well of BHI liquid medium added with a ½ serial concentrations of H_2_O_2_ (from 0.5 to 0.0004 mol/L). The final inoculum of *F. nucleatum* was 1 × 10^6^. After 48 h of incubation at 37 °C under anaerobic conditions, the plates were washed three times with phosphate-buffered saline GIBCO PBS (Thermo Fisher). The biofilm was then stained with 100 μL of 0.1% *w*/*v* of crystal violet solution (Microbial, Uta, Italy) for 10 min at 25 °C. After three washes with PBS, 200 μL of 30% *v*/*v* acetic acid was added in each well. The biofilm amount was measured by using a Thermo Scientific™ Multiskan™ FC Microplate Photometer at λ = 600 nm. The minimum biofilm inhibitory concentration (MBIC) was defined as the lowest concentration that inhibited biofilm formation, i.e., with an absorbance comparable to the negative control (sample devoid of bacteria) [16].

### 2.9. Catalase Activity

To confirm a post-transcriptional behavior for the mRNA analysis, we performed a catalase direct measurement. The method used was previously described by Yi Li et al., modified for cell culture [25]. In practice, Th medium in every plate containing the cell cultures was discarded, and after a wash with PBS, 10 mL of PBS (Sigma Aldrich, USA) was added. The cells were then scraped using the MTC™ Bio T-Shape Spreader (Merck KGaA, Darmstadt, Germany). After filtration with a 0.22 µm Millipore filter, 50 μL of the liquid was added to a solution containing 300 μL of H_2_O_2_ (3%, Masnata Elmas, Cagliari, Italy) and 1650 μL of PBS buffer (Sigma). The change in absorbance values at 240 nm was measured at T0 and after 5 min (T5) using a JASCO—V-630 spectrophotometer, following the manufacturer’s instructions (https://www.manualslib.com/brand/jasco/, accessed on 28 February 2025). Additionally, for each sample, the protein amount was measured using a Nanodrop instrument (Thermo Fisher Scientific, Silverside Road Tatnall Building, USA). The results were expressed as ΔABS at 240 nm per 1 mg of protein from experiments conducted in triplicate.

### 2.10. Statistical Analysis

Three distinct biological replicates were obtained, and quantitative data were expressed as mean ± SD. Changes in gene expression greater than 2 or less than 0.5 were considered significant. Each sample was analyzed in three separate RT-PCR runs in duplicate, and quantitative data were expressed as the mean ± SD. The Chi-square test using the social statistical program was used to evaluate the significance between different analytical groups, (*p* < 0.05) (https://www.socscistatistics.com, accessed on 28 February 2025).

## 3. Results

### 3.1. F. nucleatum Behavior in the Stressing Chamber

The stress chamber provided standard conditions for exposing the bacterium biofilm to a range of hydrogen peroxide concentrations. First, the antimicrobial test showed no sensitivity for H_2_O_2_ at the tested concentrations. In fact, the MIC value was 4000 µM, a result in accordance with Table 2 and already published paper [23].

This indicated that adding H_2_O_2_ to the chamber would cause oxidative stress without inhibiting growth. It was also plausible that the biochemical changes observed were caused by oxidative stress rather than the bacteria being in a critical vitality state. The addition of hydrogen peroxide inside the chamber resulted in consistent gas production around the cotton scaffold, which was optically evident from 100 to 1 H_2_O_2_ mM (Figure 2). This could be due to oxygen formation from H_2_O_2_ and subsequent activation of NADH, following this reaction: [NADH + H^+^ + H_2_O_2_ ⇄ NAD^+^ + 2 H_2_O + O_2_]. In this context *F. nucleatum* seems to counteract the environmental oxygenation, i.e., the increase in E_h_ [12,24,26]. At the same time, bacterial cells tolerating oxidative stress are able to increase their pathogenicity. In fact, Silva et al. (2005) asserted that the adaptation to oxidative stress by NADH might also influence the pathogenicity of *F. nucleatum*, as shown in a murine model [10].

#### 3.1.1. The VSC Production Under *F. nucleatum* Oxidative Stress

Volatile sulfur compounds (VSCs) reflect the activity of bacterial proteases. The major compounds of VSCs include hydrogen sulfide (H_2_S), methyl mercaptan (CH_3_SH), and dimethyl sulfide (CH_3_)_2_S. These substances exhibit cytotoxic activity for animal tissues [26], and are associated with the progression of various diseases, such as periodontitis and halitosis [27]. For this reason, we hypothesized that evaluating VSCs could be a direct method to assess the bacterium’s pathogenicity. The graph in Figure 3 illustrates the amounts of VSC produced at different hydrogen peroxide concentrations. Under these conditions, VSC levels were measured after approximately 24 h, revealing that dimethyl sulfide (DMS) was the most prominent metabolite in the VSC group. In contrast, H_2_S and CH_3_SH were not detectable.

Following recent research, dimethyl sulfide (DMS) has been shown to play a role in oxidative stress tissues protection through the methionine (sulfoxide reductase (MsrA) pathway. This is a key enzyme that facilities the reaction of methionine with reactive oxygen species (ROS). Using a homology modeling approach to identify an optimum substrate for the antioxidase activity, Whuan et al., (CH_3_)_2_S demonstrated that (CH_3_)_2_S is an effective substrate for catalytic oxidation. In fact, *MsrA* can bind DMS and enhanced its antioxidant capacity by facilitating the reaction of DMS with ROS (Figure 4) [27]. This protective effect can occur in two main ways: (i) protecting the oral biofilm from continuous oxidative stress in the oral cavity, and (ii) providing indirect protection against ROS formation in host cancer cells, especially those treated with some chemotherapy drugs, i.e., oxaliplatin. The *MsrA* gene is located in the *F. nucleatum* chromosome (Gene Bank accession n. CP007062.1). The role of MRSA protein in *F. nucleatum* pathogenicity is described by Scheible et al. (2022) [28]. They reported that when fusobacterial cells are exposed to hydrogen peroxide, a set of multi-gene loci is activated, such as methionine sulfoxide reductase (MsrAB), a two-component signal transduction system (ModRS), and thioredoxin (Trx)- and cytochrome C (CcdA)-like proteins. In particular, this oxidative stress defense system contributes to fusobacterial pathogenicity by enabling the attachment and penetration of colorectal target tissue while protecting against oxidative damage from immune cells [28] (Figure 4).

Following the considerations already exposed, the DMS production in *F. nucleatum* cell is strictly related to oxidative stress level, i.e., in this case, with the hydrogen peroxide amount added in the Fusobacterium stressing chamber.

#### 3.1.2. *F. nucleatum* Biofilm Formation

We investigated the ability of *F. nucleatum* cells to form biofilm after being pretreated with subinhibitory concentrations of hydrogen peroxide. In this experiment, we exposed cells to different concentrations of H_2_O_2_: (i) *F. nucleatum* cells without prior stress; (ii) *F. nucleatum* cells pre-stressed with hydrogen peroxide; and (iii) *E. coli* as a control strain

The results shown in Table 2 indicate that shifting from normal culture conditions to a growing condition with subinhibitory concentrations of H_2_O_2_ make the bacterium more resistant to the antibiofilm activity of hydrogen peroxide. At the same time, *F. nucleatum* shows an MBIC with high values in comparison to the *E. coli* control strain. This is consistent with the previous discussions [10,24]. This finding is further supported by evaluating the microbial cells adhered to HT-29 cells, as described in Section 3.2.antioxidants-14-00323-t002_Table 2Table 2H_2_O_2_-MBICs for different bacterial strains.ConditionMBIC [µmol/L]*E. coli*5 × 10^2^*F. nucleatum* ^S^2 × 10^3^*F. nucleatum* ^C^1 × 10^−3^Legend: ^S^ = *F. nucleatum* recruited from stress chamber with 100 µM H_2_O_2_; ^C^ = bacteria cell not previously stressed with H_2_O_2_.

### 3.2. HT-29 Cells Behavior During F. nucleatum Infection

The human colorectal cancer cell line HT-29 has an epithelial morphology and typically shows sensitivity to 5-fluorouracil and oxaliplatin. Additionally, this cell line appears moderately resistant to H_2_O_2_ treatment [29], which allows for the use of a wide range of hydrogen peroxide concentrations. As an initial investigation, we exposed the cell culture to different ratios of *F. nucleatum* to target HT-29 cells, with a multiplicity of infection (MOI) ranging from 1/1 to 1/1000. As reported in Table 3, using the colorimetric MTT assay, the results showed no toxicity in the MOI range from 1 to 80. For this reason, we used MOIs starting from 1/10 for the subsequent determinations.

#### 3.2.1. Bacterial Growth on HT-29 Cells

We used real-time qPCR to count the number of genomes of *F. nucleatum* attached or inside the HT-29 cell monolayer. The presence of H_2_O_2_ induced more bacterial growth of about twofold, Figure 5. For this reason, we have performed an experimental procedure to investigate the stress pattern in HT-29 cells infected with *F. nucleatum*.

#### 3.2.2. Oxidative Stress Response of HT-29 Cells Infected with *F. nucleatum*

This work underlines the role of continuous oxidative stress of low intensity on the anaerobe bacterium *F. nucleatum*. This condition must be viewed from two perspectives: increased pathogenicity and cancer host cell protection. The latter aspect was evaluated by measuring RNA targets related to oxidative stress in HT-29 cells. In this context, we measured the relative levels of HMOX1, Pri-miRNA 146a, and Pri-miRNA 155 gene expression. As previously described, HMOX1 (also known HO-1) encodes the heme oxygenase 1 protein, and its gene is likely one of the most easily induced in response to various stress stimuli, such as hypoxia, particularly oxidative stress induced by ROS [30]. According to the results presented in Figure 6A, the data show an increase in the HMOX1 expression pattern in *F. nucleatum* infected cells in the H_2_O_2_ range from 500 µM to 125 µM. Within this hydrogen peroxide concentration range, the highest gene expression levels appear in Fn-infected cells. Although the results were the highest across three different experiments, they were statistically significant (*p* < 0.01). This finding could be in line with the considerations of several authors, who suggest that this protein may induce antimicrobial effects against other microorganisms [31,32], while also being involved in promoting colorectal cancer (CRC) [32]. The role of Pri-miRNAs considered in this work is shown in Figure 6B. According to the literature, miRNA 155 is associated with a variety of tumors, including colorectal cancer associated with *F. nucleatum* [33]. Its role is in the pro-inflammatory response, cell proliferation, and chemoresistance. On other hand, miRNA 146a is involved in modulating the inflammatory process, such as tolerance to LPS [34].

The ratio S^r^ = Pri-miRNA 155/146a could indicate the stress pattern in HT-29 cells; it is plausible that a Sr > 1 suggests a high Pri-miRNA 155 value and, thus, a high-stress level compared with the control. Observing Figure 6B, the role of Pri-miRNA 155 is low in the non-infected cells treated with H_2_O_2_, while the *F. nucleatum* infection promotes the expression of Pri-miRNA 155. This could suggest the role of this bacterium in the tumor progression, event observed in CRC-infected cells. The protective effect of *F. nucleatum*-activated bacterial cells is shown in Figure 7 through three different mRNAs related to oxidative stress genes. In this case, the expression pattern is most prominent in the infected cells containing activated *F. nucleatum*. The mRNA expression levels were significantly different between the control culture and HT-29 cells infected with activated *F. nucleatum* (*p* > 0.01), as shown in Figure 7. Additionally, post-transcriptional control was performed using a catalase activity test. The results shown in Figure 8 are consistent with the observations made from the mRNA measurements.

## 4. Discussion

This work highlights the role of continuous low-intensity oxidative stress on the anaerobic bacterium *F. nucleatum*. Such a condition might frequently occur in the anaerobic biofilms present along the oral–gut axis. However, when comparing *F. nucleatum* isolates from tumors and corresponding oral strains obtained from saliva in 14 patients, Komiya et al. suggested that the Fusobacteria found in colon cancer samples originate from the oral cavity [35]. To further verify these findings, Abed et al. isolated genomic DNA from *F. nucleatum* obtained from both oral and adenocarcinoma samples from three patients. The results revealed an extremely close evolutionary relationship between the oral and corresponding tumor isolates, supporting the hypothesis that *F. nucleatum* from the oral cavity can proliferate and accumulate in colorectal cancer [36]. For these reasons, the triggering event for *F. nucleatum* stress activation, and, thus, carcinogenesis, could begin in the mouth—more specifically, in the biofilm on the tongue, where *F. nucleatum* first appears and spreads within the body [37]. The causes of this oxidative spread are likely numerous, including factors such as food intake, drug consumption, and unhealthy host habits like smoking [38,39]. An interesting aspect is related to eating habits, particularly the consumption of foods with high pro-oxidative potential. In this case, dietary components may alter the oxidative and inflammatory status of tissues. For this reason, the Dietary Inflammatory Index (DII^®^) was created to describe how inflammatory a person’s typical diet is [40,41]. Thus far, studies on the associations between diet and *F. nucleatum* have primarily focused on the gut region. For example, Narii et al. (2023) reported a study involving 212 subjects, where dietary intake and the presence of *F. nucleatum* in feces were evaluated. The authors found an inverse association between certain foods (e.g., dairy products) and the presence of *F. nucleatum* in feces [42]. Other studies have explored how colorectal cancer risk may be influenced by the relationship between *F. nucleatum* titers and pro-inflammatory diets, such as those high in red meat, refined grains, and desserts, or by antioxidant diets rich in whole grains and fiber [43,44]. Based on these findings, avoiding unhealthy dietary patterns and habits could help reduce the pathogenicity of *F. nucleatum* and potentially serve as a preventive measure against *F. nucleatum*-associated cancer. This is particularly important for young, healthy individuals, where the bacterium is primarily localized in the tongue biofilm [45]. In fact, our hypothesis is that this microorganism may invade the colorectal tract, often following a pathogenicity pattern initially activated in the oral cavity (Figure 9). However, CRC-associated strains emerge from commensal clones after specific adaptation to oncogenesis, and not all strains are associated with the neoplastic process [46,47,48].

Through the various methods described above, it emerged that *F. nucleatum* (at a 1/10 MOI) appears to have a significant role on tumor cells against oxidative stress induced by the use of hydrogen peroxide. In particular, the H_2_O_2_-induced oxidative stress determines the expression of miRNAs, in particular for Pri miRNA146a and Pri- miRNA 155, which are found to be important biomarkers involved in the mechanisms of cell growth and differentiation mechanisms. While Pri miRNA 146a has a protective role in reducing inflammation and shows changes in expression only at high concentrations of hydrogen peroxide and in the presence of the pathobiont, miRNA155 has a pro-inflammatory role and is expressed more in the presence of *F. nucleatum* associated with stress conditions. It is known that miR-155 is involved in the promotion of lung, liver, pancreas, and gastrointestinal tract cancer by repressing several molecular antineoplastics targets [49]. In contrast, Garo et al. (2021), identified microRNA-146a as an important negative regulator of colon inflammation and tumorigenesis, associated with IL-17. Mice deficient in miRNA-146a are more susceptible to colitis-associated colorectal cancer [50]. It is important to emphasize that the preclinical administration of miRNA-146a, or target inhibition of miR146a, enhances colon inflammation [51]. In this context, we measured the expression levels of Pri-miRNA 146a and Pri-miRNA 155 in HT-29 cells, stressed with H_2_O_2_ and/or infected with *F. nucleatum*. Significant changes in the two miRNAs were measured in cultures treated with a range of H_2_O_2_ [250–1000 μM]; in these conditions, the expression rate of Pri miRNA 155/Pri miRNA 146a was >1 for all H_2_O_2_ tested concentrations. The miRNA 146a–H_2_O_2_ relationship has been shown by several authors in different tissues or different cell lines. In particular, Ji et al. (2013) highlight how 146a is overregulated by H_2_O_2_-induced stress and could underregulate the expression of the SOD2 protein, decreasing both the level of SOD2 and the cell viability of the line PC12 line previously treated with H_2_O_2_ [52]. In this study, we observe that in infected cells, pri-miRNA 146a levels gradually drop compared to pri-miRNA 155 levels. This leads to an increase in SOD levels and, more importantly, an increase in catalase levels, which protects tumor cells. The data also suggest an increase in aldehyde dehydrogenase2 ALDH, an enzyme involved in the malondialdehyde (MDA) metabolism; it is probable that the upregulation of ALDH could determine a minor persistence of MDA in the cancer cells [53]. MDA is a highly reactive bifunctional compound, and it has been demonstrated that MDA cross-links the proteins and phospholipids in erythrocytes. Its increase reduced cell survival is the final effect of this process, which impairs membrane-related processes; in this context, in tumoral cells, the Fn-mediated MDA reduction could increase the tumor protection against oxidative stress. On the other hand, an important finding emerges: Although the colon adenocarcinoma cell line HT-29 at the highest concentration of H_2_O_2_ undergoes minimal mortality, it presents an innate structural resistance to oxidative stress, evidenced by the fact that, when exposed to different concentrations of H_2_O_2_, no relevant changes in terms of metabolic activity occur at the mitochondrial level. The bacterial protection against oxidative stress in HT-29 cells is also shown in Figure 6A, where in infected cells, the expression of heme oxygenase 1 (HMOX1) is significantly higher compared to non-infected cultures. These results suggest that (i) *F. nucleatum*, in comparison with other microorganisms, could be resistant against the antimicrobial activity of HMOX1, and (ii) the induction of hyperexpression of ALAD, HMXO1, CAT, and SOD could explain the role of this bacterium in promoting tumor drug protection and progression [30,31,54].

## 5. Limits of This Experimental Approach

The primary goal of this study was to investigate how HT-29 cells behaved when they were exposed to the oncogenic bacterium *F. nucleatum*. We observed a significant difference in stress protection when tumor cells were infected with *F. nucleatum* treated with H_2_O_2_. The data are interesting, but they must be confirmed with additional in vitro and in vivo experiments. First, this paper represents a mono-microbic approach, whereas in vivo, it is actually a “microbiome-mediated infection”. In practice, the infection result is due to a complex bacteria interaction, which could potentially be clarified with in vivo experiments. A second approach could involve conducting an in vitro experiment alongside a complete proteomic analysis. This would help explain how the cells behave during a real infection.

## 6. Conclusions

In conclusion, the significant impact of *F. nucleatum* on the onset and growth of CRC, as well as its involvement in drug resistance mechanisms, highlights the potential for exploiting this pathobiont as a target for preventive measures. Based on the preliminary data obtained in this study, it is possible to lay the groundwork for future preventive treatments, such as an antioxidant diet, aimed at reducing cancer incidence and progression related to *F. nucleatum*.

### Future Perspectives

We consider this work preliminary, and additional studies are necessary to confirm the perspectives raised in this manuscript. This method could help identify food compounds linked to “pathobiont inductors”, which is a new way of thinking about how *F. nucleatum* onco-virulence begins. This concept could also be intriguing in the field of oral medicine, considering that some protocols use H_2_O_2_ as a disinfectant prior to dental surgery procedures.

## Figures and Tables

**Figure 1 antioxidants-14-00323-f001:**
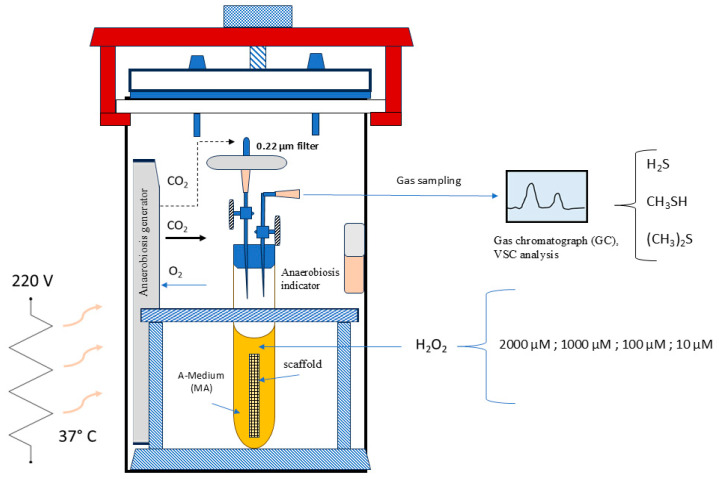
Schematic representation of the “*F. nucleatum Stressing Chamber*”.

**Figure 2 antioxidants-14-00323-f002:**
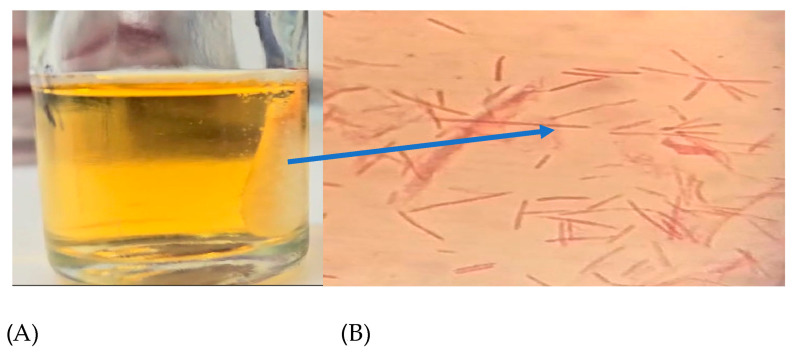
(**A**) Cotton scaffold inside a stressing chamber and (**B**) *F. nucleatum* cells confined in the scaffold; the arrow indicates the scaffold position observed by the optical microscope, with crystal violet staining at 1000X.

**Figure 3 antioxidants-14-00323-f003:**
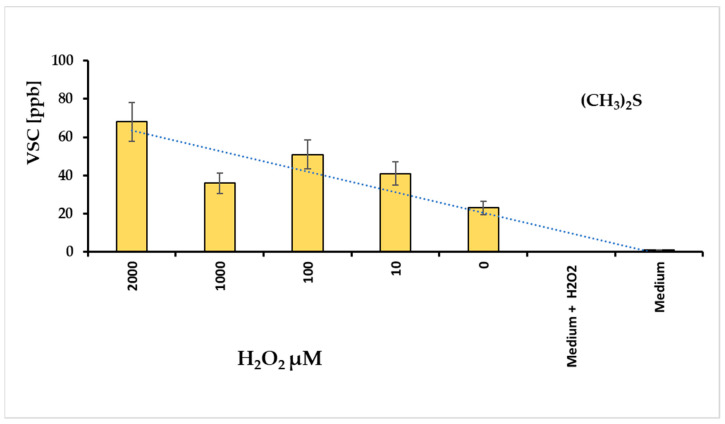
Levels of dimethyl sulfide (DMS) revealed with different concentrations of hydrogen peroxide added in a tube containing *F. nucleatum* biofilm; the culture medium added with H_2_O_2_ non revealed DMS signals.

**Figure 4 antioxidants-14-00323-f004:**
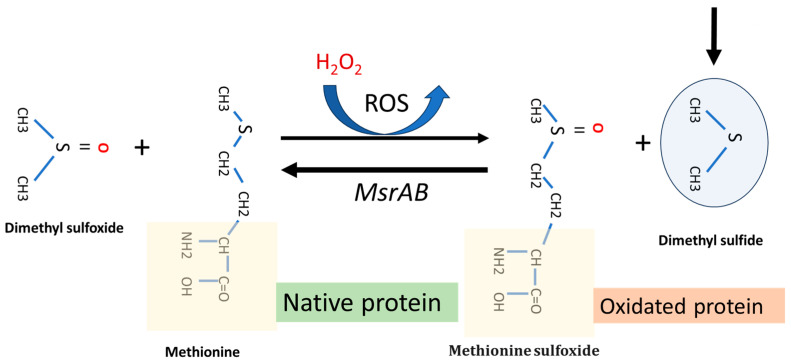
Representative function of *MsrA* gene system and dimethyl sulfide function in a *F. nucleatum* cell under H_2_O_2_-induced oxidative stress. The main arrow indicates the progress of the MsrAB mediated reaction in the presence of (CH₃)₂S.

**Figure 5 antioxidants-14-00323-f005:**
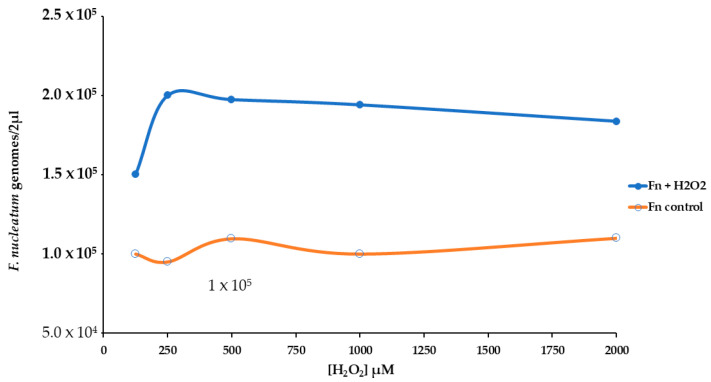
Effect of different levels of H_2_O_2_ [from 125 to 2000 µM] on the growth of *F. nucleatum* in a cell HT-29 substrate.

**Figure 6 antioxidants-14-00323-f006:**
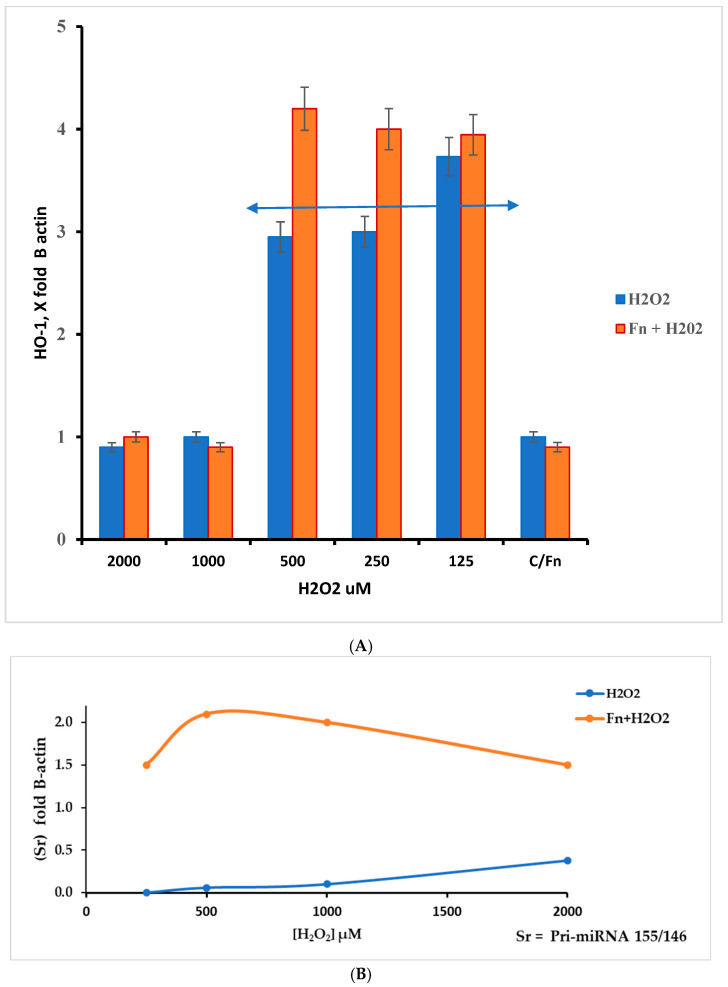
(**A**) The *HMOX1* gene responds differently to different levels of oxidative stress (H_2_O_2_) in HT-29 cells that are infected and cells that are not infected. The response window can be seen from 500 to 125 µM of H_2_O_2_. The double arrow indicates the appreciable differences in the H_2_O_2_ concentration range between infected and non-infected cells; however, *p* > 0.05. (**B**) Ratio Sr = Pri miRNA 155/146a, in control and *F. nucleatum* infected cells.

**Figure 7 antioxidants-14-00323-f007:**
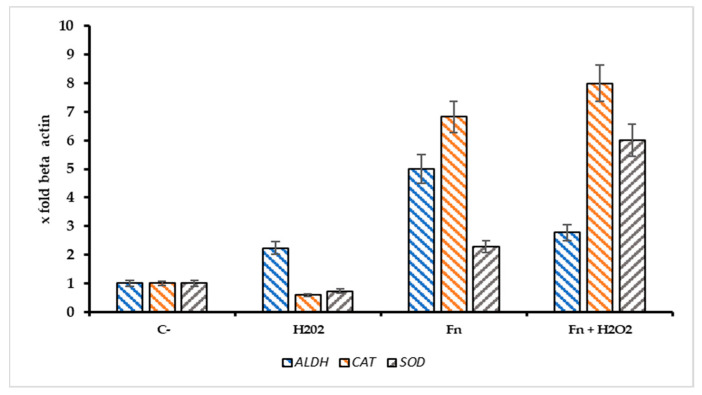
Expression pattern of *ALDH2*, *CAT*, and *SOD* genes in infected/non infected HT-29 cells with *F. nucleatum* and treated with H_2_O_2_ [500 µM]. The addition of Fn pre activated/H_2_O_2_ resulted in a significant increase of catalase mRNA (CAT) in comparison with the negative control, *p* < 0.01).

**Figure 8 antioxidants-14-00323-f008:**
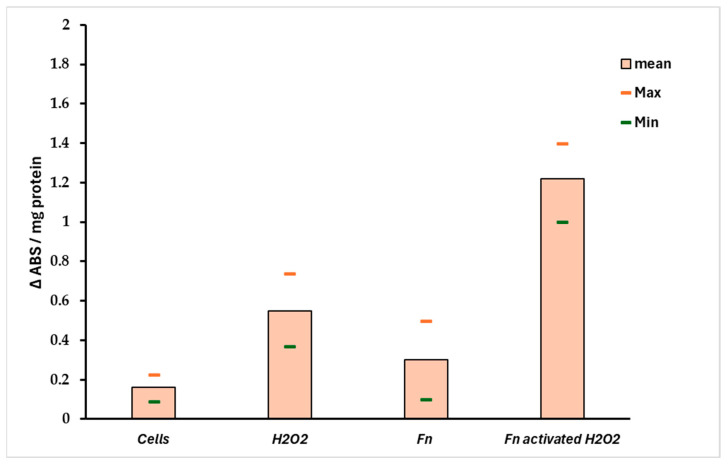
Catalase activity in infected HT-29 cells. The data support the results observed in Figure 7. A significant difference (*p* < 0.01) was observed between negative control (non-infected cells) and the HT-29 cells infected with preactivated/H_2_O_2_
*F*. *nucleatum*.

**Figure 9 antioxidants-14-00323-f009:**
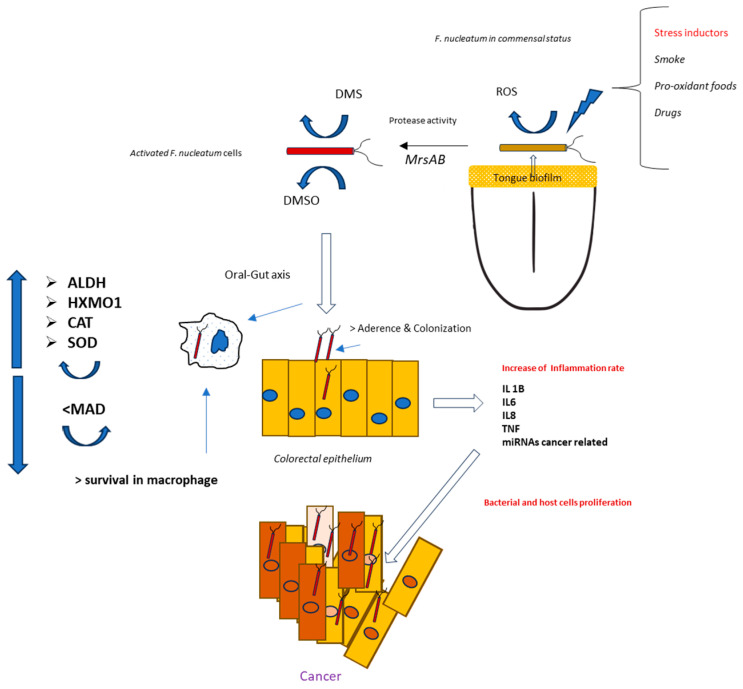
Theorical mechanisms for a pathogenic activation of *F. nucleatum* positioned in the tongue dorsum by oral stress inductors.

**Table 1 antioxidants-14-00323-t001:** PCR oligos used in this work for the relative quantitation.

Gene	Product	Forward 3′-5′	Reverse Oligo 3′-5′	Accession N.
*HO-1*	Heme oxygenase1	TACACCCGCTACCTG	TCTGGTCCTTGGTGTC	NM_002133.1
*ALDH2*	Aldehyde dehydrogenase2	GTTTACGGCCGGTCTCTTCA	GCAGCTGATGCAAGCGAAG	AH002599.2
*SOD*	Superoxide dismutase	GGTGGGCCAAAGGATGAAGA	CAAGCCAAACGACTTCCAGC	NM_000454.5
*CAT*	Catalase	ACTTCTGGAGCCTACGTCCT	GAGGGGTACTTTCCTGTGGC	NM_001752.4
*pmiR-146a*	Pri-miRNA 146a	TTTACAGGGCTGGGACAG	TCAGATCTACTCTCTCCAGG	KR606822.1
*pmiR155*	Pri-miRNA 155	AGGAAGGGGAAATCTGTG	TCATGCTTCTTTGTCATCCT	MK280370.1
*ACTB*	Beta actin	CACTGGCATCGTGATGGACT	GGCCATCTCTTGCTCGAAGT	NM_001101.5

**Table 3 antioxidants-14-00323-t003:** Assessment of HT-29 cell survival by MTT assay with different multiplicities of infection (MOI) of *F. nucleatum*.

MOI	% Cell Survival (24 h)	*F. nucleatum* N°genomes/well */***
1/1	100	1.21 × 10^5^
1/10 **	100	1.4 × 10^6^
1/20	99.9	1.89 × 10^6^
1/40	99.8	4.6 × 10^6^
1/80	99.8	9.1 × 10^6^
1/100	69.9	2.3 × 10^7^
1/1000	50.5	4.3 × 10^8^
Control	100	0

Legend: * area = 1.9 cm^2^. ** MOI rate used in these experiments. *** Total genomes consider only bacteria bound to cells.

## Data Availability

The data underlying this article are available in the article.

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
