# Peer review of "Oxidative Stress by H2O2 as a Potential Inductor in the Switch from Commensal to Pathogen in Oncogenic Bacterium Fusobacterium nucleatum"

_antioxidants, 2025, doi:10.3390/antiox14030323_

Round 1
Reviewer 1 Report
Comments:
1. Since H2O2 was the only oxidative stress used for the study, the title should be "Oxidative stress by H2O2 ..." or "H2O2-induced oxidative stress ..."
2. Needs lower dose of H2O2 in Figures 5 and 6B.
3. Does bacterium Fusobacterium nucleatum induce any specific toxins?
4. Figure 6 A needs statistical analysis.
5. Figure 7: Are other oxidative stresses such as smoking, pro-oxidant foods, and drugs studied using the same bacteria in any other similar reports?
6. Please list all abbreviations.
Comments:
1. Since H2O2 was the only oxidative stress used for the study, the title should be "Oxidative stress by H2O2 ..." or "H2O2-induced oxidative stress ..."
2. Needs lower dose of H2O2 in Figures 5 and 6B.
3. Does bacterium Fusobacterium nucleatum induce any specific toxins?
4. Figure 6 A needs statistical analysis.
5. Figure 7: Are other oxidative stresses such as smoking, pro-oxidant foods, and drugs studied using the same bacteria in any other similar reports?
6. Please list all abbreviations.
Author Response
Comment 1: Since H2O2 was the only oxidative stress used for the study, the title should be "Oxidative stress by H2O2 ..." or "H2O2-induced oxidative stress ..."
Response 1: We appreciate the reviewer’s suggestion and have changed the title.
Comment 2: Needs lower dose of H2O2 in Figures 5 and 6B.
Response 2: We thank the reviewer, we have improved to the lowest H2O2 concentration, until 62,5 µM and consequently the graphs 5 and 6 are been changed.
Comment 3: Does bacterium Fusobacterium nucleatum induce any specific toxins?
Response 3: F. nucleatum contains three prominent virulence factors: FadA, Fap2, and endotoxin LPS. Secondly are described outer membrane proteins RadD, CmpA, Aid1, FomA, serine proteases; and butyric acid.They interact strongly with the immune system and, at the same time, with the cell oxidative pattern stress to provide an environment that is conducive to the development of colorectal cancer. Because of this, the part that outside oxidant compounds play could be important for the F. nucleatum virulence factor. The manuscript provides a better description of this aspect and We thank the reviewer for pointing out this concept.
Comment 4: Figure 6 A needs statistical analysis.
Response 4: Thanks to reviewer. We have insert in the in the figure legend the p value, obtained by using Fisher exact test. The low significance has been discussed in the manuscript.
Comment 5: Figure 7: Are other oxidative stresses such as smoking, pro-oxidant foods, and drugs studied using the same bacteria in any other similar reports?
Response 5: At the present time, we have not found the same exact concept in literature. Typically, the function of certain oxidative foods is associated with host cells, not with infective bacteria, as this study hypothesized.
Comment 6: Please list all abbreviations.
Response 6: This has been done; it has been added in page 17 at the end of the manuscript.

Reviewer 2 Report
In this study, the authors studed oxidative stress using HT-29 Human colon adenocarcinoma cells and F. nucleatum subspecies animalis.
The authors used different conc of H2O2 as oxidative stress inducers and different FN MOI.
The authors measured the oxidative stress markers by qPCR
The study lacks the protein expression of oxidative stress such as Superoxide dismutase (SOD):, Catalase: , and Malondialdehyde (MDA). Therefore the findings are not convincing.
Besides the findings are not noval , previously similar findings were reported
https://academic.oup.com/femsle/article/187/1/31/474234
PMID: 37848855
Also the quality of figures and presentation is poor.
Author Response
We have remade the manuscript following the reviewer's suggestion; in fact, we have improved the figures, and we have added data regarding the gene expression of the proteins mentioned by the reviewer. In this case, the results have been described and commented on in the manuscript's text. We thank the reviewer for these suggestions.
In our opinion, our work differs significantly from the cited publication. First, the previously published paper focuses on the oxidative response of the bacterium to oxygen exposure. In contrast, our manuscript focuses on the induction of bacterium pathogenesis by a direct oxidant. For this reason, we have also used a cell infection model, a system not represented in the publication by Diaz et al. However, the cited publication was crucial in the field of bacteria's response to internal oxidative stress and has been cited and commented on in the text.

Reviewer 3 Report
Review for the manuscript
Antioxidants (ISSN 2076-3921)
Manuscript ID: antioxidants-3449287
Title:
Oxidative stress as a potential inductor in the switch: commensal-pathogen in oncogenic bacterium Fusobacterium nucleatum
Special Issue:
Role of Reactive Oxygen Species (ROS) in Tumor Microenvironment Modulation
Dear Editor,
Thank you for the invitation to review for ANTIOXIDANTS. I have some comments and suggestions regarding this manuscript before it can be accepted for publication.
OVERALL COMMENTS
In this study, the authors say that Fusobacterium nucleatum is a pathobiont that plays a dual role as a commensal and a pathogen and is closely linked to colorectal cancer due to its potential involvement in tumor progression and resistance to chemotherapy. The mechanism for its transformation to pathogen from commensal remains unknown. Based on this statement, the authors intended to investigate the role of oxidative status as an initiatory factor for changing the bacterium pathogenicity profile. Their results showed that the bacterial colonization rate, dimethyl sulfide production, and pri-miRNA 155 led to increased levels by using stressed bacteria, suggesting a main pathogenic function for these strains.
This is a topic that can be an important aid in increasing knowledge about Fusobacterium nucleatum aspects in progression and resistance to treatment of colorectal cancer.
Minor general comments:
In many parts of the text, we find Fusobacterium nucleatum or F. nucleatum. After the first time you use F. nucleatum, only use like this. Sometimes the name of the bacteria is in italics, and sometimes, it is not. Please use a standardization.
TITLE
The title is “Oxidative stress as a potential inductor in the switch: commen- 2 sal-pathogen in oncogenic bacterium Fusobacterium nucleatum”.
I suggest changing the title for “Oxidative stress as a potential inductor in the switch commensal to pathogen in oncogenic bacterium Fusobacterium nucleatum. Please note that the name of the bacteria should be in italics.
ABSTRACT
This section is fine. I just suggest that the authors review some grammar mistakes in this section along with the entire manuscript.
KEYWORDS
The Keywords are adequate.
INTRODUCTION
This section is fine. However, the references are not new. Can authors include newer references in this section and along the entire text?
There are many very good studies that the authors can find in PUBMED that elucidate many points regarding Fusobacterium nucleatum and colorectal cancer. Please, check them. Only in 2024 there are almost 30 articles regarding Fusobacterium nucleatum and colorectal cancer.
METHODS
This section was well described.
Figure 1 is very nice.
RESULTS
· In line 237 we can find a section named “Results and Discussion”. However, it seems to me that this should be only “Results” since we find another section named “Discussion”.
· Indeed, in the section, we find discussion points. But after we find a Discussion section separated. I suggest including all the Discussion in only one section.
· The Figures are very nice.
DISCUSSION
· 1- As suggested above (all the Discussion of the study´s findings should be in only one topic).
· 2- I suggest adding newer references.
· 3- Should Figure 7 be in this section?
CONCLUSION
The conclusion is short but is fine.
I suggest including a separate section with the limitations of this study and Future Perspectives.
REFERENCES
As mentioned above.
Please check grammar and references as I suggested in my comments and suggestions.
Author Response
Comment 1: In many parts of the text, we find Fusobacterium nucleatum or F. nucleatum. After the first time you use F. nucleatum, only use like this. Sometimes the name of the bacteria is in italics, and sometimes, it is not. Please use a standardization.
Response 1: We appreciate the reviewer’s suggestion and we have corrected the bacteria names.
Comment 2: TITLE The title is “Oxidative stress as a potential inductor in the switch: commen- 2 sal-pathogen in oncogenic bacterium Fusobacterium nucleatum”. I suggest changing the title for “Oxidative stress as a potential inductor in the switch commensal to pathogen in oncogenic bacterium Fusobacterium nucleatum. Please note that the name of the bacteria should be in italics.
Response 2: Thank you for pointing this out. We have modified the title as you suggested. Moreover, we added 'H2O2-induced oxidative stress' to better explain the type of stress used in our study.
Comment 3: ABSTRACT This section is fine. I just suggest that the authors review some grammar mistakes in this section along with the entire manuscript.
Response 3: Thank you. We have reviewed and corrected the English grammar throughout the entire manuscript
Comment 4: KEYWORDS The Keywords are adequate.
Response 4: We appreciate the reviewer’s comment.
Comment 5: INTRODUCTION This section is fine. However, the references are not new. Can authors include newer references in this section and along the entire text? There are many very good studies that the authors can find in PUBMED that elucidate many points regarding Fusobacterium nucleatum and colorectal cancer. Please, check them. Only in 2024 there are almost 30 articles regarding Fusobacterium nucleatum and colorectal cancer.
Response 5: Thank you. We conducted a more recent search on major scientific platforms (PubMed, Google Scholar, etc.) and tried to integrate the bibliography with newer studies.
Comment 6: METHODS This section was well described. Figure 1 is very nice.
Response 6: Thanks to the reviewer’s for this comment.
Comment 7: RESULTS: - In line 237 we can find a section named “Results and Discussion”. However, it seems to me that this should be only “Results” since we find another section named “Discussion”.
- Indeed, in the section, we find discussion points. But after we find a Discussion section separated. I suggest including all the Discussion in only one section.
- The Figures are very nice.
Response 7. The typing mistakes have been corrected; we thank the reviewer for pointing this out.
Comment 8: DISCUSSION: 1- As suggested above (all the Discussion of the study´s findings should be in only one topic). 2- I suggest adding newer references. 3- Should Figure 7 be in this section?
Response 8. (1) We have unified the discussion session as the reviewer suggested; (2) at the same time, we have added the most recent bibliography; and (3) our intention was to insert a final image to describe the research hypothesis, renamed now as Figure 8. However, we are available to change the figure position in the manuscript.
Comment 9: CONCLUSION The conclusion is short but is fine. I suggest including a separate section with the limitations of this study and Future Perspectives.
Response 9. This has been done; we have added the new session at the end of the manuscript.
Comment 10: REFERENCES As mentioned above.
Response 10: We have modified the references by adding newer ones.

Round 2
Reviewer 1 Report
No more comments.
No more comments.
Author Response
Thank you.
Reviewer 2 Report
The authors did not perform protein levels of the requested targets, but mRNA level.
Still the conclusion is not solid
The authors did not perform protein levels of the requested targets, but mRNA level.
Still the conclusion is not solid
Author Response
Thank you for your valuable comments and advice. The main goal of this paper was to examine HT-9 cells infected with a strain of F. nucleatum before being treated or not with different amounts of hydrogen peroxide. The primary evaluations included cell viability, bacterial intracellular growth, and oxidative gene expression. While it is true that we only focused on the pre-transcriptional aspect, it is also true that many authors only report mRNA results, especially when they also describe cellular behavior (e.g., vitality index), such as how cells react to different amounts of Hâ‚‚Oâ‚‚ and how easily bacteria can grow.
For example, here is a brief list of similar approaches:
- https://doi.org/10.1093/jnci/87.2.117 – Hypoxia expression pattern in HT-29 cells, expression patterns, and vitality/apoptotic cells.
- https://doi.org/10.3390/ijms222212286 – HT-29 cells treated with nanocarrier-encapsulated camptothecin (only expressome analysis).
- https://doi.org/10.1242/jcs.106.3.771 – Differential expression of human mucin genes, expression patterns, and the phenotypic presence of mucus droplets in HT-29 cells.
- https://doi.org/10.1038/sj.bjc.6602864 – Hypoxia-conditioned cell lines derived from HCT116 and HT-29, VEGF and HIF-1α cDNAs mRNA analysis.
- And others.
These papers focused on gene expression along with morphological or functional aspects (e.g., cell growth).
Additionally, to address the experimental data on mRNAs from a post-transcriptional perspective, we have measured catalase activity under the same experimental conditions already described for mRNA detection. This methodology and the corresponding results have been added to the manuscript.
Finally, we have included a discussion of the experimental limitations at the end of the manuscript.
We hope that these considerations and the modifications to the manuscript are sufficient for the acceptance of the paper.
